# What Indicators Are Shaping China's National World-Class High-Tech Zones? Constructing a Feature Indicator System Based on Machine Learning

**Sida Feng [1], Hyunseok Park [2,\*] and Fang Han [3,\*]**

1    College of Economics and Management, Beijing University of Chemical Technology, Beijing 100029, China; sidafeng@buct.edu.cn
2    Department of Information Systems, Han Yang University, Seoul 02581, Republic of Korea
3    National Science Library, Chinese Academy of Sciences, Beijing 100190, China
\*    Correspondence: hp@hanyang.ac.kr (H.P.); hanfang@mail.las.ac.cn (F.H.)

**Abstract:** China's high-tech parks have significant effects on driving national ecological innovation. Among them, ten world-class high-tech parks represent the highest level of development in China's high-tech industry. Understanding the development characteristics of national world-class high-tech parks is of great significance for guiding the construction of other parks and achieving the high-quality development of parks. Based on the evaluation data of over 200 indicators of national high-tech parks from 2013 to 2017, this study used the XGBoost classic machine learning algorithm to select the characteristic indicators of national world-class high-tech parks and establish an evaluation indicator system, and it identified four primary indicators of the world-class high-tech parks, including innovation development, enterprise development, international development, and economic development. The indicators cover 30 important sub-indicators and highlight the importance of innovation resource input indicators, such as "use of technology activity funding from government departments", "full-time equivalent of R&D personnel", and "financial technology expenditure in high-tech parks". Compared to the expert analysis, the application of the machine learning method in the evaluation of national high-tech parks improves the efficiency of selecting important indicators and makes the selection results more objective. The results of this research provide a reference value for guiding and promoting national high-tech parks to become world-class parks.

**Keywords:** national high-tech park; world class; feature indicator system; machine learning; high-quality development

## 1. Introduction

With the rapid development of technology and the economy, China's national high-tech zones have become powerful engines, driving the transformation of regional economic structure and growth [1–3]. The history of China's national high-tech zones could be traced back to 1988, when the first high-tech zone "Zhongguancun" was established in Beijing. Its development history can be summarized into the following three periods.

Before 2000, the main focus of high-tech zone construction was to gather production factors, and the main objectives were to quickly establish an industrial foundation and achieve economic scale. The second entrepreneurial stage (2001–2010) emphasized the introduction of technological elements, including the introduction of research and development institutions as well as scientific and educational resources. The third entrepreneurial stage (2011–present) signifies a new phase of "comprehensive innovation" in the construction of national high-tech zones. With a focus on comprehensive innovation, high-tech zone construction must consider all elements conducive to innovation and create an environment favorable for innovation and independent research and development. This phase has led

high-tech zones to be transformed towards comprehensive development in the "innovation economic ecosystem".

National high-tech zones play important roles in the innovation behavior of knowledge-intensive service enterprises [4]. Currently, the number of China's national high-tech zones is more than 170. And, according to the report "National High-Tech Zone" released by the Ministry of Science and Technology of China, in 2017, the GDP of the total 156 national high-tech zones reached CNY 9.52 trillion, accounting for 11.5% of national GDP. The ratio of R&D expenditure to GDP was 7.09%, which is 3.3-times the national average. The number of patent applications and authorizations accounted for 20.8% of the national total. The number of authorized invention patents per 10,000 employees was more than ten-times the national average. National high-tech zones have become significant carriers for promoting technological progress and enhancing independent innovation, as well as powerful engines driving the transformation of regional economic structure and economic growth.

However, the development of national high-tech zones in China exhibits a significant imbalance. In the "National high-tech zone" reports, China's national high-tech zones are classified into three tiers: world class, innovative parks, and other parks. Currently, there are ten high-tech zones in China that have been approved to join the sequence of world-class park construction, including Zhongguancun, Shanghai Zhangjiang, Shenzhen, Wuhan East Lake, Xi'an, Chengdu, Hangzhou, Suzhou Industry, Hefei, and Guangzhou high-tech zones. The development of world-class parks far surpasses that of other parks. According to the report, in terms of the total number of enterprises, the number of high-growth enterprises and listed companies in these ten world-class high-tech zones accounted for 45.3%, 57.8%, and 35.7%, respectively, of the total in all national high-tech zones. Regarding economic indicators, the operating income, industrial added value, export volume, and net profit of these ten world-class high-tech zones accounted for nearly half of the total in national high-tech zones. In terms of innovation indicators, the number of personnel with a bachelor's degree or above, R&D expenditure, and number of invention patents in these ten world-class high-tech zones accounted for over 50% of the total in national high-tech zones. These ten world-class high-tech zones have become the pioneers in developing high-tech industries in China, adjusting industrial structures, driving the transformation of traditional industries, and enhancing international competitiveness.

On the other side, most other high-tech zones in China have poor performance. Thus, it is of great significance to fully grasp the development characteristics of these world-class parks, understand the significant differences between other parks and world-class parks, and provide guidance for the future development of other parks in order to achieve high-quality development for national high-tech zones and even the whole country.

The evaluation of national high-tech zones has always been an important part of the research on high-tech zones. The construction of an evaluation index system plays an important guiding role in the development of high-tech zones. In recent years, the number of national high-tech zones and the related evaluation data have been rapidly increasing [5]. Faced with the enormous and complex evaluation data, it is important to carry out efficient and accurate data analysis. Many studies have been conducted on the construction of comprehensive and innovation evaluation index system for high-tech zones, most of which are based on expert evaluation, statistical evaluation, and other methods [6]. These methods have high professional requirements and strong subjectivity, and they are easily influenced by the subjective factors of the evaluators, thus lacking scientific rigor.

With continuous development, more and more applications of machine learning in evaluation have emerged [7–9]. Compared with previous methods, machine learning algorithms are more objective, not influenced by the subjectivity of evaluators, and have strong adaptability and high accuracy in various scenarios. Therefore, machine learning has achieved great success in evaluation research in many fields. However, in the current research, there are few studies that use machine learning algorithms for high-tech zone evaluation and even fewer studies exploring the application of machine learning in the ex-

ploration of world-class park characteristic index systems. Apparently, the main limitation can be due to a lack of data. Evidence regarding the performance of high-tech zones in a nation is very limited.

As an initial exploration of the application of machine learning in high-tech zone evaluation research, this study introduces the widely used XGBoost algorithm in machine learning. Considering the availability of data, this paper chooses all China's national high-tech zones during the period from 2013 to 2017 as a sample. Based on the relevant statistical data of the Torch Center of the Ministry of Science and Technology, this study analyzes the significant differences between world-class high-tech zones and other national high-tech zones in each year and constructs a characteristic index system for world-class high-tech zones based on the weighted average of five annual feature indexes. This provides guidance for the development of other parks towards world-class parks.

## 2. Literature Review

### 2.1. Theoretical Research on China's National High-Tech Zones

There has been extensive and rich theoretical research on high-tech zones. Many scholars have conducted research on the innovation capabilities, industrial development, and innovation systems of China's national high-tech zones.

First, many scholars have analyzed the current situation and influencing factors of the development of innovation capabilities in national high-tech zones. Based on panel data from 251 cities from 2004 to 2016, Li et al. [10] investigated the impact and the mechanism of China's HTZ upgrading policy on urban innovation underlying this effect, and they found that the place-based upgrading policy significantly improves the innovation level of cities. Jiang et al. [11] used the stochastic frontier approach to study the efficiency change in national high-tech zones and found that variables related to the international division of labor promoted production efficiency improvements in high-tech zones, while pure technical efficiency caused total factor productivity degradation. The quality of capital accumulation and management system efficiency were identified as the main reasons for the superiority of Eastern high-tech zones over Central and Western ones. Zhang et al. [12] explored the influence of foreign direct investment quantity and quality on the low-carbon development of Chinese science and technology parks based on data from 2011 to 2018. Wang et al. [13] empirically analyzed the impact of scientific and technological innovation policies on the innovation efficiency of high-technology industrial parks via linear regression and qualitative comparative analysis.

In terms of innovation system research, Sun and Wang [14] investigated and analyzed the construction and development of the innovation systems in national high-tech zones, summing up the problems existing in China's high-tech industry cluster innovation system. Cai et al. [15] used the innovation ecology to construct an evaluation system for the ecological quality of the innovation system in high-tech zones and used on-site investigations of 15 high-tech zones in Hubei Province to extract six main factors of the innovation ecology system through factor analysis; in terms of influencing factors research, Hu et al. [16] empirically examined the scale-increasing effect of knowledge production in high-tech zones using panel regression analysis and found that knowledge stock and human capital play a significant role in the knowledge production process in China's high-tech zones. Zeng et al. [17] used the three-stage DEA-Tobit analysis method to estimate the innovation efficiency of national high-tech zones and found that the degree of capital aggregation, enterprise clustering, talent clustering, and industrial agglomeration within the high-tech zones are the main factors causing regional efficiency differences. Utilizing the SBM-VRS and Tobit model and based on 54 of China's national high-tech zones from 2007 to 2010, Yang et al. [18] empirically analyzed the impacts of FDI on the operation and innovation performance of China's national high-tech zones, and they found that, in general, FDI has a positive effect on improvements in the operation performance of high-tech zones but has no significant effect on their innovation performance.

In terms of industrial development, Hu and Nie [19] proved that the formation of industrial agglomeration in high-tech parks is driven by the inherent dynamics of high-tech industries, which promote the concentration of high-tech industries in specific regions. On the other hand, high-tech parks have an attractive effect on high-tech industrial activities, attracting various economic elements to gather together. Liu and Luo [20] summarized the development experience of advanced international science and technology parks and their influencing factors. In addition, taking empirical evidence from the biopharma cluster in the Greater Boston Area and based on the data from 2012–2017, Ferretti et al. [21] analyzed whether and how the micro-geographical proximity influences the formation of the relationships of venture capital (VC) deals, intellectual property (IP) transfer agreements, and R&D strategic alliances.

The aforementioned research indicates that although Chinese national high-tech zones have become national innovation demonstration zones, they still face shortcomings, such as generally low innovation efficiency, lack of industrial agglomeration effects, and extremely uneven development of parks. How to identify the characteristics of industrial development in world-class high-tech zones based on improvements in the innovation development system is of great guidance and reference significance in promoting the overall innovation capabilities of national high-tech zones and fully leveraging their driving force.

### 2.2. Research on the Evaluation of High-Tech Zones

Many scholars have studied the evaluation of high-tech zones. For example, based on the qualitative analysis of science and technology parks, Markusen and Venables [22] conducted quantitative research on science and technology parks through index decomposition. Alessandro [23] analyzed the promotion of innovation by high-tech parks using the example of Moglino, Russia. Bigliardi et al. [24] assessed science parks' performance based on an analysis of selected Italian parks and found the important role of environment context, the stakeholder's commitment, and the life cycle of the science park. Nosratabadi et al. [25] applied a fuzzy expert system to the evaluation of science and technology parks. Albahari et al. [26] provided a theoretical framework of national science park's evaluation and applied it to the Italian and Spanish systems.

In China, there are several influential evaluation systems of national high-tech zones formulated by the government. For example, the evaluation system laid down by the Ministry of Science and Technology reveals the comprehensive innovation abilities of all the national high-tech zones. "Zhongguancun Index" and "Zhangjiang Index" evaluate the two tech zones' innovation ability, respectively. Chinese scholars have also conducted a large amount of work in evaluating high-tech zones. For example, Guo et al. [27] established an evaluation system for high-tech zone competitiveness from four dimensions: technological innovation, capital support, entrepreneurial environment, and development performance. Fan [28] established an evaluation system for the technological innovation capacity of high-tech zones, which includes 16 indicators, such as total assets, R&D expenses, and R&D personnel, through a comprehensive study of technological innovation processes and the use of expert methods and discriminant analysis. Chen and Wang [29], starting from economic transformation, constructed an evaluation system for innovation transformation indicators from four perspectives: industrial clusters, innovation input, innovation output, and innovation environment. Liu et al. [30] constructed an indicator system for the high-quality development level of national high-tech zones. They used the entropy method to measure the high-quality development level of national high-tech zones in China from 2013 to 2018.

However, there is limited research on the evaluation system for China's world-class high-tech zones. A search on CNKI (China National Knowledge Infrastructure) yielded only two relevant articles. Hu [31] used principal component analysis and SPSS17.0 to evaluate the capital supply and demand capacity of six national autonomous innovation demonstration zones striving to become world class. Wu [32] systematically reviewed the development theories and evaluation indicators of high-tech parks both domestically and

internationally. They proposed a comprehensive evaluation indicator system for world-class parks based on the balanced scorecard. They partially applied it to the empirical investigation of creating a world-class park in Suzhou Industrial Park, identified problems, and provided recommendations.

According to statistics and analysis, there are over 10 methods used in the construction of evaluation systems for high-tech zones. Commonly used methods include the analytic hierarchy process (AHP), expert scoring, factor analysis, multi-level grey evaluation, and data envelopment analysis (DEA). In recent years, with the development of machine learning algorithms, some scholars have applied machine learning in evaluations. For example, Li [9] used the Random Forest method to construct a risk indicator system for real estate projects and verified the correctness of the Random Forest algorithm in the real estate project risk evaluation model. Tang and Li [7] constructed a personal credit assessment model based on the Bagging ensemble learning algorithm and tested the model using data from a credit card customer database of a branch of a commercial bank in China. The results showed that ensemble learning algorithms can significantly improve the prediction accuracy of decision trees. However, there is limited research on the use of machine learning in evaluating high-tech zones. Zhang [8] applied different machine learning algorithms, such as Random Forest, Gradient Boosting, and support vector machines (SVMs) in the evaluation of innovation capacity and indicator prediction of Inner Mongolia's High-Tech Zone.

In comparison, machine learning algorithms are more objective compared to other evaluation methods as they do not require human intervention, thus avoiding subjective bias. Traditional evaluation methods, such as factor analysis and analytic hierarchy process, are limited to linear functions, while the impact of indicators on evaluation values is often not limited to linear relationships [8]. Machine learning algorithms, on the other hand, are nonlinear models, which can more accurately reflect the relationship between indicators and comprehensive evaluation results.

## 3. Evaluation Data and Related Algorithms

### 3.1. Data Sources

When constructing the indicator system for world-class parks, it is necessary to combine the development characteristics of these parks with targeted approaches. At the same time, some basic principles should be followed, such as scientificity, directionality, and operability. Scientificity means that the indicators should be important factors that influence the innovative capabilities of world-class parks. The indicators must also possess completeness and be systematic. Directionality means that the purpose of the evaluation is to analyze the development characteristics of world-class parks and guide and promote the development of other parks. Evaluators need to have a systematic understanding of the development direction and process of high-tech zones, which should be reflected through the evaluation indicator system and its evolution. Furthermore, the establishment of the indicator system is for practical application, so each indicator should be feasible and operational. Therefore, it is advisable to use simple indicators to reflect the characteristics of world-class parks.

Based on the analysis of the principles for constructing the indicator system mentioned above, the dataset used in this study comes from two categories of data compiled by the Torch Center of the Ministry of Science and Technology of China from 2013 to 2017, which covers all national high-tech zones. The aim is to ensure the integrity and objectivity of the dataset.

The first category of data includes "enterprise"-related data of national high-tech zones. Enterprises are the main force driving high-tech development, occupying a dominant position in technological research and development, product design, and market promotion. The data related to "enterprises" include the number of enterprises in each high-tech zone, the average and end-of-year number of employees, authorized patents, scientific and technological personnel, institutional funding expenditure, and the number of personnel with bachelor's, master's, and doctoral degrees. The second category of data includes

"park"-related data from national high-tech zones. The development of high-tech zones is a complex system process that requires the participation and mutual development of governments, universities, research institutions, and financial institutions. The park data include the total production value of the park, the area of the park, total revenue and expenditure of the high-tech zone, the number of research institutes, and the number of various universities. Table 1 shows part of the configuration detail data in 2017.

**Table 1.** Statistical summary of indicators in China's national high-tech zones.

| | Enterprise Indicators | | | | Park Indicators | | | |
|---|---|---|---|---|---|---|---|---|
| | Number of Enterprise | Full-Time Equivalent Number of R&D Personnel | Gross Industrial Output Value (Thousand Yuan) | Number of Employees | Total Fiscal Revenue (Thousand Yuan) | Number of Universities | New Industrial Technology R&D Institutions | Number of Patent Applications in 2017 |
| Max | 22,013 | 182,450.2334 | 1,108,438,915 | 2,620,437 | 75,380,980 | 47 | 136 | 74,368 |
| Min | 23 | 7.083333333 | 3,214,506.7 | 5017 | 0 | 0 | 0 | 17 |
| Mean | 11,018 | 10,500 | 131,833,827.6 | 125,410.18 | 84,344,747.92 | 5.53 | 7.7 | 37,192.5 |
| Standard Variance | 1883.18 | 21,446.31455 | 168,250,294.4 | 238,861.6828 | 131,858,59.58 | 7.98 | 16.46 | 8679.91 |

From 2013 to 2017, the number of national high-tech zones increased from 115 to 157 (including Suzhou Industrial Park). The number of national high-tech zone indicators collected by the Torch Center of the Ministry of Science and Technology increased from 246 to 350 (Table 2). In this study, for each year, all the collected statistics were organized and analyzed to obtain the indicator values of the characteristics of world-class parks for each year. Based on the average weights of the characteristic indicators, an indicator system for world-class parks was constructed.

**Table 2.** Number of Chinese national high-tech zones and evaluation indicators.

| Year | Number of National High Tech Zones | Statistical Indicators |
|---|---|---|
| 2013 | 115 | 246 |
| 2014 | 116 | 232 |
| 2015 | 147 | 232 |
| 2016 | 147 | 244 |
| 2017 | 157 | 350 |

*3.2. Data Standardization Processing*

Due to the different dimensions of evaluation indicators, standardization processing is required for each evaluation method. The standardization function includes linear and non-linear methods. The most commonly used methods include min–max standardization, log function transformation, and arctan function transformation. In this study, the widely used min–max standardization method, also known as range standardization, is employed to linearly transform the original data, making the results fall within the [0, 1] range. Assuming that the number of indicators is *m*, and the number of the sample is *n* in the year *t*, $X_{ij}$ is the value of the *j*th indicator of the *i*th sample. $Z_{ij}$ is the normalized value of $X_{ij}$. $X_{j-max}$ and $X_{j\ min}$ represent the maximum and minimum value of the *j*th indicator in the *n* sample, respectively. The transformation function is shown as follows:

$$Z_{ij} = \frac{X_{ij} - X_{j-min}}{X_{i-max} - X_{j-min}} \tag{1}$$

Generally, standardization functions are designed for indicators where a larger value is considered to be better. However, there are cases where smaller values are considered

better (in such cases, the monotonicity needs to be changed by taking the reciprocal or negating the value). In this study, almost all indicators can be considered as "the larger, the better" type and, hence, can be directly transformed. Taking the year 2017 as an example, the standardized values of various indicators for national high-tech zones are shown in Table 3.

**Table 3.** Standardized value of evaluation indicators of national high-tech zones in 2017.

| National High-Tech Zones | Number of Enterprises | R&D Personnel Full-Time Equivalent | Internal Expenditure on R&D Funds | Park Added Value | Industrial Added Value | Operating Revenue |
|---|---|---|---|---|---|---|
| Zhongguancun | 1.000 | 1.000 | 1.000 | 1.000 | 0.959 | 1.000 |
| Shenzhen | 0.094 | 0.465 | 0.521 | 0.195 | 0.228 | 0.137 |
| Hangzhou | 0.091 | 0.290 | 0.315 | 0.147 | 0.258 | 0.103 |
| Zhangjiang | 0.242 | 0.498 | 0.615 | 0.518 | 1.000 | 0.363 |
| Wuhan | 0.138 | 0.455 | 0.401 | 0.251 | 0.418 | 0.226 |
| Suzhou industrial park | 0.115 | 0.320 | 0.234 | 0.140 | 0.358 | 0.093 |
| Chengdu | 0.086 | 0.201 | 0.154 | 0.147 | 0.331 | 0.111 |
| Hefei | 0.058 | 0.184 | 0.185 | 0.159 | 0.350 | 0.086 |
| Guangzhou | 0.173 | 0.450 | 0.341 | 0.159 | 0.310 | 0.134 |
| Xi'an | 0.180 | 0.315 | 0.363 | 0.296 | 0.632 | 0.211 |

*3.3. XGBoost Algorithm and Application*

Machine learning, fundamentally, is similar to statistical regression analysis. Most machine learning algorithms are regression algorithms that seek the functional relationship between independent and dependent variables. Unlike other parametric tests, the biggest advantage of using machine learning algorithms in evaluations is that they do not make assumptions about the functional relationship between independent and dependent variables, thus avoiding human involvement and subjective biases.

There are many machine learning algorithms. Commonly used ones include artificial neural networks, support vector machines, adaptive spline regression, and various Bagging and Boosting techniques. Among them, ensemble learning techniques, like Bagging and Boosting, have been a hot topic in machine learning research in recent years. They attempt to combine multiple learning models to reduce fitting errors and improve the prediction accuracy of individual models [33]. The main representative algorithm for Bagging is Random Forest, and for Boosting, there are algorithms like Adaboost, GBDT, and XGBoost. Currently, XGBoost allows the user to define the loss function, further increasing the model's generalization ability. In addition, a series of strategies, such as the greedy algorithm for finding the structure of added trees and the loss function and regularization term in the loss function, makes XGBoost more accurate in prediction compared to other methods, thus gaining wide application [34].

Based on this, in this study, the national high-tech zones are classified into "world-class parks" and "other parks" categories. The XGB classifier from the XGBoost is used to train and fit the standardized indicator data from 2013 to 2017, and the accuracy rate, precision rate, recall rate, and F1 score [35] are used to evaluate the performance of the model; the grid search method is used for tuning hyperparameters to improve the models' performance in order to determine the feature indicators that have a significant influence on distinguishing excellent parks from other parks. The importance of each feature indicator is then ranked based on their weights. The frame of the structure is as follows (Figure 1):

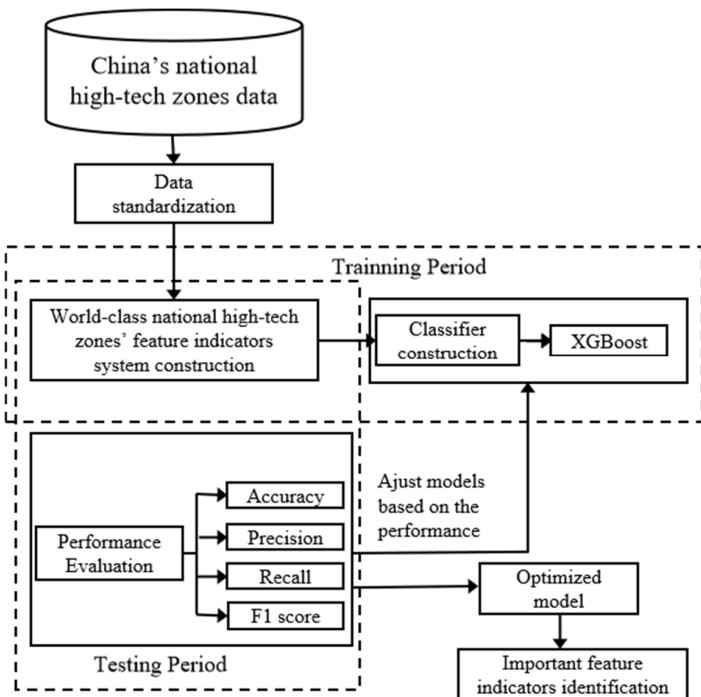

**Figure 1.** The identification of feature indicators of world-class high-tech zones based on XGBoost.

## 4. Results

### 4.1. Selection of Feature Indicators

Based on the XGBoost algorithm, we randomly selected 80% of the data for training, and the other 20% of the data were utilized for performance testing. The parameters of this model were optimized by utilizing the grid search method based on the performance. Taking 2017, for example, the parameters were set as shown in Figure 2. And the mean of the parameters was as follows:

eta: the parameter for learning rate (also known as the step size).

n_estimators: the number of decision trees (iterations).

max_depth: the maximum depth of each tree, controlling its complexity and preventing overfitting.

reg_alpha: the weight of L1 regularization.

reg_lambda: the weight of L2 regularization.

min_child_weight: the minimum sum of sample weights required at a child node during tree growth.

learning_rate: the rate controlling the weight update magnitude for each tree.

colsample_bytree: the fraction of features used for training each tree.

As a result, the accuracy, precision, recall, and F1 score for experiments in 5 years (2013–2017) are all over 0.75, which indicates the validity and feasibility of XGboost in selecting the feature indicators of world-class high-tech zones.

```
XGBClassifier(eta=0.25619670282702833, n_estimators=3000, max_depth=5, reg_alpha=0, reg_lambda=1,
              min_child_weight=1, gamma=2, learning_rate=0.47093835312978727, colsample_bytree=0.18,
              verbosity=0)]
```

**Figure 2.** The parameter set in XGBoost model for the year 2017 in this study.

Then, the results for selecting feature indicators for excellent parks show significant differences in weights among the evaluation indicators across different years, among the over 200 evaluation indicators. Over 80% of the indicators have small weights (<0.01), while there are 20–30 indicators that have a significant contribution to distinguishing excellent parks from other parks. Taking the year 2017 as an example, approximately 20 indicators

have contribution weights greater than 0.01 (see Figure 3). Among them, indicators, such as park value added, number of graduate employees, number of valid patents, government funding for scientific and technological activities, and number of patent transfers and licenses, have higher weights. Five indicators contribute over 70% towards distinguishing excellent parks from other parks, indicating that they are significant indicators of the advantages of excellent parks.

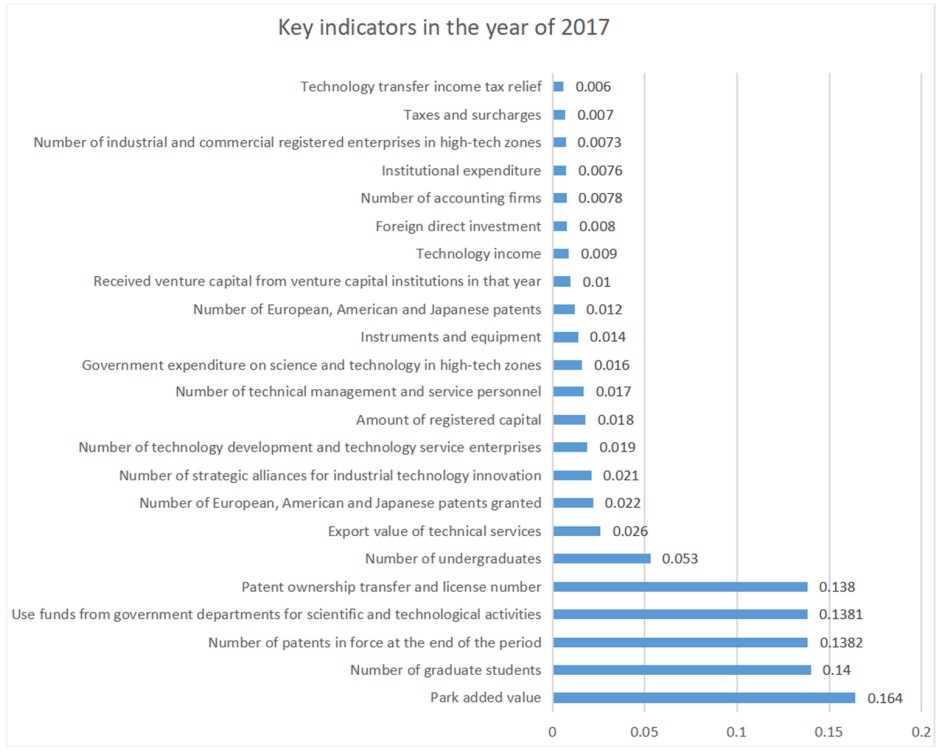

**Figure 3.** The contribution degree of the evaluation indicators in 2017.

Based on the above analysis, this article selected the top 20 indicators ranked by weight from 2013 to 2017 as the feature indicators for excellent parks in each year (Table 4).

**Table 4.** The characteristic indicators of ten world-class national high-tech zones (2013–2017).

| Rank of Indicator | 2013 | 2014 | 2015 | 2016 | 2017 |
|---|---|---|---|---|---|
| 1 | Number of high-tech enterprises | Valid patent | Authorized domestic invention patents | Authorized domestic invention patents | Park added value |
| 2 | Software copyright | High tech enterprises | R&D personnel | Income tax | Graduate practitioner |
| 3 | Total personnel of R&D institutions | Total agency personnel | Technical service export | R&D personnel | Number of patents in force at the end of the period |
| 4 | R&D personnel | Master's degree or above returnee | Overseas students start businesses | Institutional expenditure | Use funds for scientific and technological activities of government departments |
| 5 | Master's degree or above returnee | Income tax | Total amount of taxes and fees actually paid | Trademark registered in that year | Patent ownership transfer and licensing |

**Table 4.** *Cont.*

| Rank of Indicator | 2013 | 2014 | 2015 | 2016 | 2017 |
|---|---|---|---|---|---|
| 6 | People who have returned from studying abroad | Technical service export | High level | Number of listed enterprises | Undergraduate practitioners |
| 7 | Technical consulting and service income | Operating profit | Apply for domestic invention patents | Overseas students start businesses | Technical service export |
| 8 | Number of overseas marketing agencies established by enterprises | Number of overseas marketing service agencies established by enterprises | Receiving commissioned research and development income | Non-operating income | Authorizes European, American and Japanese patents |
| 9 | Total volume of technical contract transactions | Value-added tax | Park added value | Invention patent | Industrial technology innovation strategic alliance |
| 10 | The software copyright was obtained in that year | Have a registered trademark | Paid-in capital (equity) | Ending mechanism number | Technology development and technology service enterprises |
| 11 | Number of listed enterprises | Overseas students start businesses | Full-time science and technology activity staff | Technical service export | Registered capital |
| 12 | Number of overseas technology research and development institutions established by enterprises | Funds for scientific and technological activities of government departments shall be used for internal scientific and technological activities of enterprises | Year-end owner's equity | Authorized invention patent | Science and technology management and service personnel |
| 13 | Number of mechanisms | Thousand Talents Plan | Subsidy income | Park added value | Government expenditure on science and technology in high-tech zones |
| 14 | Research and development plus deduction for income tax | Research and development plus deduction for income tax | Income from technology transfer | Doctoral practitioner | Expenditure on instruments and equipment for scientific and technological activities |
| 15 | Form international standards | Technology transfer mechanism | Funds for scientific and technological activities of government departments shall be used for internal scientific and technological activities of enterprises | Asset impairment loss | It has European, American and Japanese patents |
| 16 | Employees with master's degree | That year received venture capital | Number of overseas marketing service agencies | Foreign direct investment | That year received venture capital |

**Table 4.** *Cont.*

| Rank of Indicator | 2013 | 2014 | 2015 | 2016 | 2017 |
|---|---|---|---|---|---|
| 17 | Funds for scientific and technological activities of government departments shall be used for internal scientific and technological activities of enterprises | Park added value | Number of enterprises | Expenditures for scientific and technological activities entrusted to foreign entities | Technology income |
| 18 | Total export | Patent ownership transfer and licensing | Expenditures for scientific and technological activities carried out by entrusted units | Government expenditure on science and technology in high-tech zones | Foreign direct investment |
| 19 | Trademark registered in that year | Industrial and commercial registered enterprise | That year received venture capital | Provincial and above technology transfer demonstration institutions | Accounting firm |
| 20 | National qualified product inspection and testing institutions | Apply for domestic invention patents | Apply for European, American and Japanese patents | Introduce foreign experts | Institutional expenditure |

### 4.2. The Feature Indicator System of World-Class National High-Tech Zones

Based on the analysis of the top indicators for excellent parks in five years, the differences between the top ten parks and other parks mainly include indicators, such as "number of high-tech enterprises", "number of R&D personnel", "number of highly educated employees", and innovation output. In addition, there has been an increasing number of indicators representing international innovation capabilities, making internationalization an increasingly significant advantage for top parks.

Based on the selection of feature indicators from 2013 to 2017, this article establishes an indicator system to grasp the development advantages of top parks from both macro and micro perspectives. The system selects the top 30 feature indicators with the highest average weights over the five years and classifies them into four primary indicators: innovation development, enterprise development, international development, and economic development. In addition, to maintain the independence of indicator meanings and eliminate redundancy, some highly correlated indicators are processed. For example, for the indicators of "number of employees with a bachelor's degree or above" and "number of employees with a master's degree or above", only the former is retained. The sum of the average weights of the selected 30 feature indicators after processing is 0.821, indicating that these indicators can effectively represent the development advantages of top parks. Among them, "number of high-tech enterprises", "number of valid patents", and "added value of the park" rank in the top three of the weighted ranking of feature indicators. After normalizing the weights of the 30 feature indicators, the weights of each indicator in the evaluation system can be seen in Table 5.

**Table 5.** The characteristic index system of ten world-class national high-tech zones.

| One-Leve Indicator (Weight) | Category | Characteristic Indicator | Weight |
| --- | --- | --- | --- |
| Innovation and development (0.431) | Resource | R&D personnel full-time equivalent | 0.047 |
| | | Number of employees with a bachelor's degree or above | 0.013 |
| | | Number of institutions at the end of the period | 0.013 |
| | | Investment amount from venture capital firms obtained during the year | 0.010 |
| | Input | Use of funds for scientific activities from government departments | 0.049 |
| | | Technological expenditure by high-tech zones | 0.043 |
| | | Expenditure on technology activities entrusted to external units | 0.039 |
| | | Income tax reduction for R&D additional deductions | 0.011 |
| | Achievement | Number of valid patents at the end of the period | 0.117 |
| | | Number of patent ownership transfers and licenses | 0.038 |
| | | Software copyrights | 0.017 |
| | | Number of domestic invention patent applications | 0.014 |
| | | Total amount of technology contract transactions | 0.010 |
| | | Obtained software copyrights during the year | 0.010 |
| Business development (0.214) | Scale | Number of high-tech enterprises | 0.121 |
| | | Number of listed companies | 0.017 |
| | | Registered trademarks during the year | 0.037 |
| | Economic indicators | Expenditure on institutional funds | 0.018 |
| | | Actual amount of taxes paid | 0.015 |
| | | Equity at the end of the year | 0.007 |
| Internationalization development (0.198) | Go global | Number of overseas technology research and development institutions established by enterprises | 0.032 |
| | | Amount of direct overseas investment | 0.048 |
| | | Technology service exports | 0.040 |
| | | Number of overseas marketing service institutions | 0.024 |
| | Introduce | Enterprises founded by overseas students | 0.029 |
| | | Returned overseas students | 0.024 |
| Economic development (0.157) | Park | Park value-added | 0.061 |
| | Enterprise | Non-operating income | 0.044 |
| | | Paid-in capital | 0.043 |
| | | Operating profit | 0.010 |

Table 5 shows that compared to other parks, the top ten parks have prominent advantages in innovation development. The primary indicator of "innovation development" has the highest weight at 0.431, indicating strong independent innovation capabilities of top parks. Additionally, top parks also have significant advantages in "enterprise development", "internationalization development", and "economic development", highlighting their strong ability in cultivating and operating businesses, as well as their good economic benefits. They are also characterized by their willingness to venture overseas and actively participate in international cooperation and competition.

Based on analysis of the indicators of ten world-class parks and their actual development experiences, strengthening the agglomeration of innovation resources and creating a favorable environment for innovation and entrepreneurship seem crucial for improving the performance of the other high-tech zones. For example, talents are the core competitiveness for the Zhongguancun high-tech zone [36], and the park has attracted a large number of scientists, entrepreneurs, investors, and various professional service talents who complement each other. In addition, in recent years, various world-class parks have developed innovative and entrepreneurial platforms in the form of new research and development institutions and heavy-incubation spaces.

According to the indicator system, actively cultivating high-tech enterprises and promoting high-quality industrial development are other key factors for becoming world-

class high-tech zones. A previous study claimed that forming high-tech enterprise clusters could improve the innovation performance of enterprises [37]. Wuhan Donghu High-tech Zone, for example, with the optoelectronic information industry as its focus, has been continuously building the world-famous "Optics Valley" and gathering a high-end and complete industrial chain. Hangzhou High-tech Zone has taken the lead in areas, such as internet innovation and entrepreneurship, cultivating e-commerce and other new forms of rural economy to improve industrial economic efficiency, and comprehensively creating an "Internet Plus" entrepreneurial center. Enterprises must be supported to go global and promote cross-border open innovation. According to our result, "International development" is becoming an increasingly prominent characteristic and advantage of world-class parks. For example, Suzhou Industrial Park, as an exemplary cooperation between China and foreign countries, has adopted Singapore's experience in economic and public management, enabling the investment environment and other aspects of the park to be aligned with the world [38]. In addition, Suzhou Industrial Park continues to deepen Sino–Singapore cooperation, and initiatives, such as the Belt and Road, financial openness, and service trade, have become new cooperation highlights of the park.

However, it is important to note that China's ten world-class parks still have big gaps compared with the world's excellent parks in some respects. For example, according to the Silicon Valley Index 2018, the Silicon Valley region received a significant increase in venture capital investment, reaching USD 14 billion, while Zhongguancun has only received USD 0.89 billion, which is significantly lower than Silicon Valley. The per capita income in Silicon Valley reached USD 93,707, and the highest average salary of employees in China's national high-tech zone, Shanghai Zizhu High-tech Zone, is USD 33,677.4 per year.

## 5. Conclusions

This study applied the XGBoost algorithm, a machine learning ensemble algorithm, to analyze the indicators of ten world-class parks from 2013 to 2017. Based on the 20 key differentiating indicators identified for each year, a world-class park indicator system was constructed to analyze the development characteristics of world-class parks from both macro and micro perspectives.

This study is meaningful for understanding the development characteristics and advantages of world-class parks and promoting the high-quality development of national high-tech zones. However, there are still some limitations. First of all, although quantitative analysis can improve the objectivity of evaluation to some extent, it may have limitations in practicality. Combining quantitative analysis with qualitative analysis, such as expert evaluation to further improve the indicator system and enhance its practicality and guiding significance, is one of the future research directions. Second, due to the data accessibility, the data of China's national tech-zones for recent years as well as the data of famous parks of other countries have not been included in this work. In addition, more machine learning models could be utilized to analyze the feature indicators of world-class high-tech zones. Thus, we can select the model with higher classification accuracy.

**Author Contributions:** Conceptualization, S.F. and F.H.; methodology, S.F. and F.H; software, S.F. and F.H.; validation, S.F.; formal analysis, S.F., F.H. and H.P.; investigation, F.H.; resources, F.H.; data curation, F.H.; writing—original draft preparation, S.F. and F.H.; writing—review and editing, F.H. and H.P.; visualization, S.F.; supervision, F.H. and H.P.; funding acquisition, S.F. and H.P. All authors have read and agreed to the published version of the manuscript.

**Funding:** This research is supported by grants from the National Natural Science Foundation of China (Grant No.42101176), the National Key Research and Development Program of China (2022YFC2105400), the Funds for world-class Discipline Construction (XK1802-5), and the National Research Foundation of Korea (NRF), grant funded by the Korean government (MSIT) (No. 2023R1A2C2006962).

**Institutional Review Board Statement:** Not applicable.

**Informed Consent Statement:** Not applicable.

**Data Availability Statement:** Data available on request from the corresponding authors.

**Conflicts of Interest:** The authors declare no conflict of interest.

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
