# Peer review of "What Indicators Are Shaping China’s National World-Class High-Tech Zones? Constructing a Feature Indicator System Based on Machine Learning"

_applsci, doi:10.3390/app131910690_

Round 1

Reviewer 1 Report

Journal

Applied Science 

Manuscript ID

applsci-2545468

The authors uses the XGBoost classic machine-learning algorithm to select the characteristic indicators of national first-class high-tech parks and establish an evaluation indicator system. The following points need to be addressed before publication of the article:

  1. The data has been used from 2013 to 2017, what will be the credibility of such data in 2023.
  2. The authors uses XGBoost classic machine-learning algorithm, what is the reason of using this algorithm and how works better against other such algorithms.
  3. The authors claim that The XGBoost algorithm performs well in selecting the characteristic indicators of hightech parks. What they means “performs well”, what are the parameters, what was the criteria and how they validated their claim?
  4. How authors claim that the proposed algorithm improve efficiency if the proposed work is novel approach as stated by the authors.
  5. There should be some sentences in end of abstract that shows the performance evaluation of proposed model and how much accuracy in terms of percentage has been improved by the authors.
  6. Reference style is not as per the journal of Applied sciences, please revise it accordingly.
  7. A proper motivation is missing in introduction section, the authors should provide some research motivation in respect of Chinese National First-Class High-Tech Parks through benchmark research papers.
  8. Introduction section is not enough to support the stance of the authors, it is recommended to extend in some extent.
  9. In the related work the latest and relevant work is missing, authors are recommended to include some latest and relevant benchmark articles from 2021 to 2023.
  10. Sections number are not proper, please revise it.
  11. Heading of section 2 should be refined and it should contain some text before starting of next subsection.
  12. The authors written a statement, “expected to increase to over 170 by 2023”, it is already 2023, how they justified this claim?
  13. The research in proposed domain in the article is silent from 2018 to 2022, it seems that the authors conducted research in 2017 and then now they submitted for applications. They have to revise relevant sections to justify this research in 2023.
  14.  Figure 1 is not properly formatted, it is blurred and it makes confusions as there is no proper explanation of the Figure 1, the authors should revise it as per recommendations.
  15. There is no proposed model/framework, how the authors conducted research, what steps they have used and what was the environment they conducted research.
  16. It should be better if the authors provide categories in a table with some configuration details.
  17. Equation 1 lacks the iteration, whether it is the equation derived by the authors or it was previously used, if it is derived by the authors then justify its working in your proposed algorithm and if it was previously used then provide references and also provide justification how it is applicable in your proposed algorithm.
  18. It should be better if some graphical results of the proposed algorithm is given compared with some benchmark algorithm from literature review.
  19. Conclusion is not written professionally, it should be revised with some outcomes of the research and limitations of the research followed by future directions.
  20. Section 5 is totally unpredictable at current stage, the authors should justify it and whether this is discussion on results or literature. Such discussion should be supported by proper citations.  

 Minor editing of English language required

Author Response

Please see the attachment. Thanks so much for your help and efforts!

Reviewer 2 Report

The title of the article should be made more interesting, from stating that it is researching feature indicators of excellence through machine learning technique, to questioning what feature indicators of such excellence are. As for specifying that a machine learning technique is used, it may be given as a subtitle instead.

National High-tech Zones descriptions should not be treated as a literature review but should instead be an introductory part in the Introduction section.

A synthesis of feature indicators of excellence, especially in the broader picture than in the case of China, should be added to outline the framework of the study. However, it would allow this research to have a wider scope than just the Chinese parks and be linked to more international research.

Discussions with research and case studies of other countries should be added. It will make this research more interesting and useful for readers who are not only interested in the case of China, but also the excellence of other similar organizations in other countries.

Many of the references are quite old and should be updated to strengthen the framework of the study as well as sharpen the debate.

Author Response

Please see the attachment, thanks so much for your valuable comments!

Reviewer 3 Report

Dear authors,

I would like to thank the authors for his manuscript. This is an interesting and applicable topic. There are some 

 Please consider my comments below; which I hope will be of help to the authors.

The method was not described in the manuscript. it is essential to explain the proposed approach or even the applied scheme using pseudo-code or flowchart. 

Most of the references are out of date. Please add more recent related work.

Kind regards

Author Response

Please see the attachment. Thanks so much for your efforts!

Round 2

Reviewer 1 Report

I appreciate the efforts of the authors to address my comments. However, I still have some concerns to further address before publication of the manuscript. They also not provided the highlighted version that reflects changes made by them. 

1. The first comment is addressed by the authors partially, they do explain that what is the reason of using this algorithm and how works better against other such algorithms.

The authors claim that The XGBoost algorithm performs well in selecting the characteristic indicators of hightech parks. What they means performs well, what are the parameters, what was the criteria and how they validated their claim?

How authors claim that the proposed algorithm improve efficiency if the proposed work is novel approach as stated by the authors.

There should be some sentences in end of abstract that shows the performance evaluation of proposed model and how much accuracy in terms of percentage has been improved by the authors.

2. The authors explain that they used accuracy rate, precision rate, recall rate, and F1 score (Forman, 2003) to evaluate the performance of the model, how they do no provided the details how they used it with which configurations . Secondly if there is no existing benchmark as per their stance then how they claim that they improve the accuracy, precision rate, recall rate or F1 score. It is good to show your work graphically based on these parameters with various configurations. 

Minor editing of English language required

Author Response

Thanks so much for your valuable suggestions. We are sorry that we inadvertently failed to upload the highlighted document last time. We have made careful revisions based on each of your suggestions.

        First of all, the focus of this study is an application-oriented  research, so we have only applied XGBoost as the algorithm (this limitation has been emphasized at the end of the paper). We have removed the previous claim of "perform good". In fact, Since there were very few literature for comparison using the same data, we only elaborated on the objectivity and efficiency of using machine learning method (compared with the expert evaluation method).

     Secondly, we have added xgboost's setting parameters in the revised paper (in the beginning of Section 4.1).

      Thank you very much for your suggestions!

Reviewer 3 Report

Thank you, all the issues are responced.

Author Response

Thanks for your valuable suggestions again!